# Time-restricted eating and supervised exercise for improving hepatic steatosis and cardiometabolic health in adults with obesity: protocol for the TEMPUS randomised controlled trial

Alba Camacho-Cardenosa [1], Antonio Clavero-Jimeno [1], Juan J Martin-Olmedo [2], Francisco Amaro-Gahete [3,4,5], Rocío Cupeiro [1,6], María Trinidad González Cejudo [7], Patricia Virginia García Pérez [8], Carlos Hernández-Martínez [1], Raquel Sevilla-Lorente [2], Alejandro De-la-O [1,5], Alejandro López-Vázquez [1], Marcos Molina-Fernandez [1], Almudena Carneiro-Barrera [9], Federico Garcia [3,10,11], Alba Rodríguez-Nogales [3,12,13], Julio Juan Gálvez Peralta [3,12,14], Rafael Cabeza [15], José L Martín-Rodríguez [8], Araceli Muñoz-Garach [3,5,16], Manuel Muñoz-Torres [3,17,18,19], Idoia Labayen [20,21], Jonatan R Ruiz [1,3,5]

AC-C and AC-J are joint first authors.

For numbered affiliations see end of article.

**Correspondence to**
Dr Jonatan R Ruiz;
ruizj@ugr.es and
Dr Alba Camacho-Cardenosa;
acamachocardenos@ugr.es

## ABSTRACT

**Introduction** Metabolic dysfunction-associated steatotic liver disease is a major public health problem considering its high prevalence and its strong association with extrahepatic diseases. Implementing strategies based on an intermittent fasting approach and supervised exercise may mitigate the risks. This study aims to investigate the effects of a 12-week time-restricted eating (TRE) intervention combined with a supervised exercise intervention, compared with TRE or supervised exercise alone and with a usual-care control group, on hepatic fat (primary outcome) and cardiometabolic health (secondary outcomes) in adults with obesity.

**Methods and analysis** An anticipated 184 adults with obesity (50% women) will be recruited from Granada (south of Spain) for this parallel-group, randomised controlled trial (TEMPUS). Participants will be randomly designated to usual care, TRE alone, supervised exercise alone or TRE combined with supervised exercise, using a parallel design with a 1:1:1:1 allocation ratio. The TRE and TRE combined with supervised exercise groups will select an 8-hour eating window before the intervention and will maintain it over the intervention. The exercise alone and TRE combined with exercise groups will perform 24 sessions (2 sessions per week+walking intervention) of supervised exercise combining resistance and aerobic high-intensity interval training. All participants will receive nutritional counselling throughout the intervention. The primary outcome is change from baseline to 12 weeks in hepatic fat; secondary outcomes include measures of cardiometabolic health.

**Ethics and dissemination** This study was approved by Granada Provincial Research Ethics Committee (CEI Granada—0365-N-23). All participants will be asked to provide written informed consent. The findings will be disseminated in scientific journals and at international scientific conferences.

**Trial registration number** NCT05897073.

---

## STRENGTHS AND LIMITATIONS OF THIS STUDY

⇒ TEMPUS aims to determine whether time-restricted eating (TRE) combined with exercise is superior to TRE or exercise alone in reducing hepatic fat and improving cardiometabolic health in adults with obesity.

⇒ This study is uniquely designed to discern and evaluate these effects separately in both men and women with obesity.

⇒ TEMPUS will quantify the persistence of the effects of a 12-week TRE and exercise intervention on hepatic fat and cardiometabolic health in the long-term, with a follow-up at 12 months.

⇒ The study is limited to adults with obesity aged 25–65 years, and its results cannot be extended to older populations or to people with other pathologies.

⇒ Future studies should also consider assessment of an increased length of the intervention.

---

## INTRODUCTION

Obesity prevalence has steadily increased up to reach epidemic proportions and affecting around 603.7 million adults worldwide.[1] The excess of triglycerides in the body is usually stored, apart from the subcutaneous adipose tissue (SAT), in other organs and tissues

that are not otherwise designed for adipose storage.[2] This process is known as ectopic fat deposition and may include organs and tissues such as the liver, pancreas or skeletal muscle. Excessive accumulation of triglycerides in hepatocytes results in hepatic steatosis, a condition considered one of the diagnostic criteria (along with other metabolic dysregulatory factors) for metabolic dysfunction-associated steatotic liver disease (MASLD).[3 4] MASLD (which replaces non-alcoholic fatty liver disease, NAFLD) is a major public health problem considering its elevated prevalence (nearly 90% of adults with overweight/obesity) and its strong association with extrahepatic diseases.[3 5] Therefore, implementing strategies to reduce hepatic steatosis in individuals with obesity may be a potential approach to mitigate/reduce the risk of liver dysfunction and cardiometabolic diseases.[6]

Traditionally, low-calorie diets have been shown to be an effective strategy to reduce body weight and hepatic steatosis and, in turn, improve cardiometabolic health.[7] However, energy-restricted approaches are still not a standard public health strategy due to their lack of long-term sustainability and undesirable metabolic adaptations, which certainly lead to weight regain even in highly motivated patients.[8] Time-restricted eating (TRE) is a recently emerged intermittent fasting approach, which has the potential to maximise the extensively reported beneficial metabolic effects of the energy intake restriction.[9] TRE aims to maintain a consistent daily cycle of feeding (within a limited time window during ≤10 hours) and fasting (≥14 hours) to support healthy/consistent circadian rhythms.[10] Irregular eating patterns and eating over an extended period of time may disrupt circadian rhythms and, thus, increase the risk of obesity and hepatic fat accumulation.[11] Remarkably, recent studies in mice have concluded that TRE effectively reduces hepatic steatosis and improves cardiometabolic health,[12] mainly through improved insulin sensitivity; yet, whether this strategy is similarly effective in humans remains still unclear.

Along with nutritional strategies, exercise has demonstrated its efficacy at reducing hepatic steatosis and at improving cardiometabolic health in humans.[13–15] Furthermore, preliminary evidence has highlighted that the combination of TRE and exercise may normalise glucose homeostasis and improve lipid profile in women with overweight or obesity.[16] Nevertheless, the differential effects of TRE combined with exercise and TRE or exercise alone on hepatic steatosis and cardiometabolic markers remain unknown.

Although promising, most preliminary pilot trials examining the effects of TRE combined with exercise in humans have important limitations: (1) the duration is shorter than 3 months[16–19] which may be insufficient to induce substantial changes in cardiometabolic health[20]; (2) the outcomes assessed were either body mass index (BMI) as a surrogate marker of obesity or bioelectrical impedance analysis which are not able to accurately assess hepatic or other ectopic fat depots[18]; (3) the majority of clinical trials have been focused solely on men,[17–19] or

have included an unevenness sample of men and women which limits the possibility to understand the important sex dimorphism in MASLD development; (4) the published studies are limited to trained individuals[17–19] and have small sample size and, thus, have limited statistical power; (5) the previous work does not include follow-up of the participants to understand the maintenance of intervention effects; and, more importantly, (6) have not studied or reported potential mechanisms through which TRE combined with exercise may result in health benefits. Therefore, as all these shortcomings limit generality and preclude to establish firm conclusions, a new approach is required to successfully translate findings into the community and clinical setting.

The overall aim of the TEMPUS randomised controlled trial is to investigate the effects of a 12-week TRE combined with a supervised exercise intervention, compared with TRE or supervised exercise alone and with a usual care (UC) control group, on hepatic fat (primary outcome) and cardiometabolic health (secondary outcomes) in men and women with obesity.

## METHODS AND ANALYSIS
### Study design
The TEMPUS study is a randomised controlled trial ( ClinicalTrials.gov, NCT05897073) with a four-arm parallel design. This protocol is reported following Standard Protocol Items: Recommendations for Interventional Trials (SPIRIT) guidelines[21] and the results will be reported following the Consolidated Standards of Reporting Trials (CONSORT) guidelines. Consented participants will be randomly assigned to one of the four groups: UC, TRE alone (TRE), supervised exercise alone (Exercise), or TRE combined with supervised exercise (TRE+Exercise) group. The study recruited adults with obesity in Granada, a region located in the southern region of Spain. Figure 1 is a patient flow diagram from recruitment to randomisation.

### Participants and eligibility criteria
The study will include both men and women (50%), with a BMI ranging from 30 to <40 kg/m$^2$, aged between 25 and 65 years, and with a habitual eating window of ≥11 hours.

Detailed criteria for inclusion and exclusion can be found in box 1. The screening phase, as shown in figure 1, will involve assessing participants' medical history and vital signs to determine their eligibility for the study. During the physical examination, any existing conditions at the time and any pre-existing medical conditions will be thoroughly documented.

### Recruitment and screening
Recruitment of potential participants will be conducted through (1) newspaper advertisements, (2) the Endocrinology and Nutrition Department of the San Cecilio' and Virgen de las Nieves' University Hospitals of Granada and (3) the community of the University of Granada. A

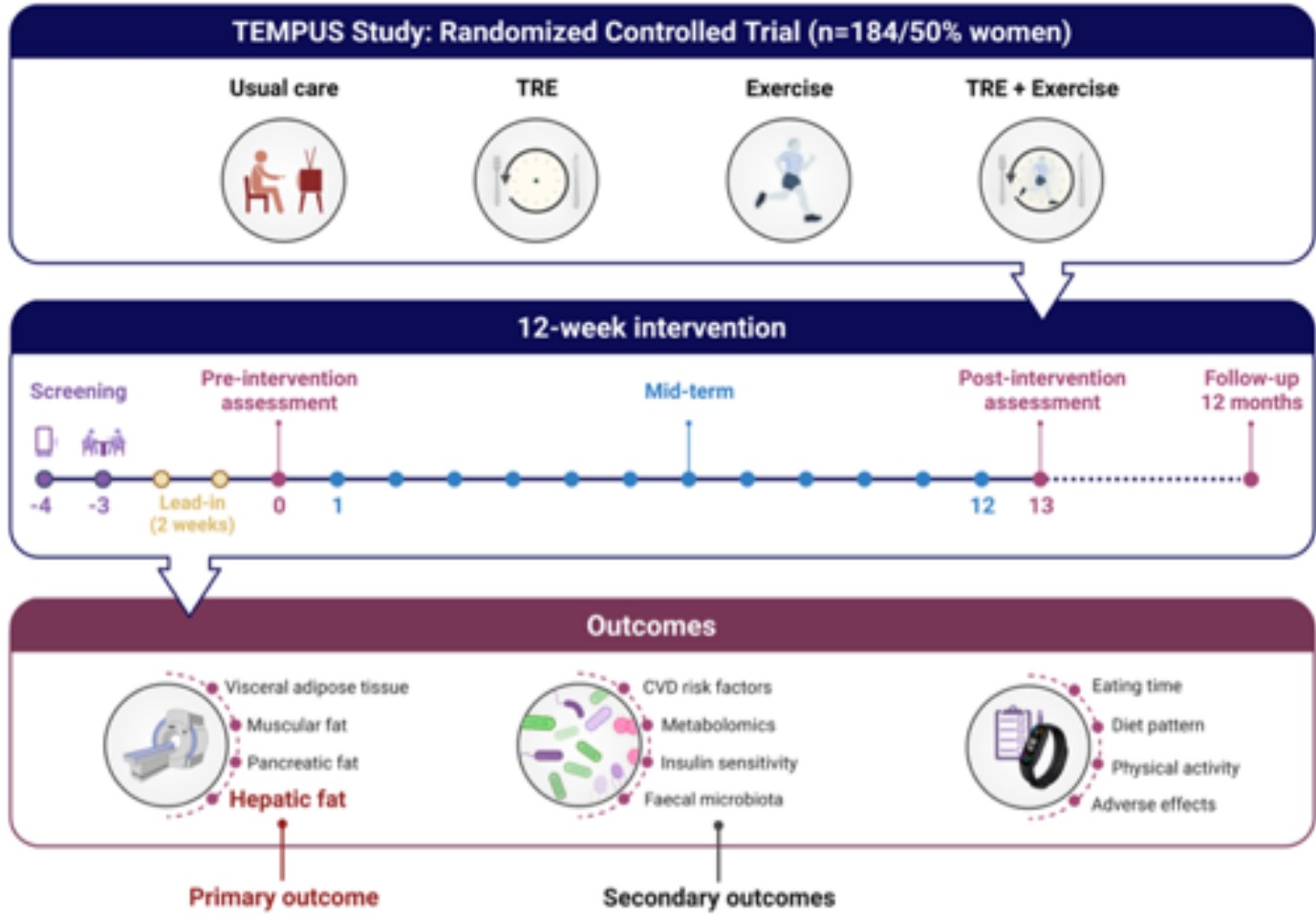

**Figure 1.** TEMPUS project design. CVD: cardiovascular diseases.

**Figure 1** TEMPUS project design. TRE, time-restricted eating; CVD, cardiovascular diseases.

prescreening process will be implemented with an online form as well as via telephone interviews to assess eligibility of potential participants and engage them in the study. Subsequently, the medical team will conduct telephone interviews, to review the patients' medical records and evaluate their medical history, verifying compliance with the inclusion and exclusion criteria. Potential participants who meet the criteria during the prescreening phase will be provided with oral and written information about the study and will be scheduled for the first evaluation visit. During this session, oral and written informed consent will be obtained from potential participants (online supplemental patient consent form 1), and measurements of body weight and height will be recorded. Additionally, participants will perform an incremental exercise test on a treadmill, supervised by the sports medicine staff from the Centro Andaluz de Medicina del Deporte, to determine their aptitude for exercising. During the test and at rest, ECG, blood pressure and capillary blood lactate measurements will be obtained.

## Sample size

Based on previous findings from a recent trial on the combination of alternate day fasting and exercise on hepatic fat content,[22] we anticipate approximately 5.0% reduction in hepatic fat content in the TRE+Exercise group, 2.5% in the TRE group, 2.5% in the exercise group and no significant change in this outcome in the usual-care group. Assuming a pre–post correlation of 0.8 and an SD of 6 points in the main outcome, we estimate a medium effect size of 0.45. To detect this effect size as statistically significant in a one-way analysis of variance with $\alpha=0.05$ and a power of 0.8, a minimum of 19 patients per group is required. Accounting for subgroup analyses based on sex and a maximum drop-out rate of 20%, we will aim to recruit 46 participants for each trial group, resulting in a total sample size of 184 participants, with 92 women included.

To ensure a balanced representation of both sexes and maintain an adequate sample size, we will implement several strategies:

– Recruitment process: A specific recruitment process will be used to aim for an equal enrolment of men and women. For each woman recruited, we will encourage her to invite one man to participate in the study. This approach has been successfully employed in our previous intervention studies and has proven

## Box 1 Inclusion and exclusion criteria

**Inclusion criteria**
⇒ Aged 25–65 years.
⇒ Body mass index 30 and <40 kg/m$^2$.
⇒ Weight stability (within 3% of screening weight) for >2 months prior to study entry.
⇒ Habitual eating window ≥11 hours.

**Exclusion criteria**
⇒ History of a major adverse cardiovascular event (eg, acute myocardial infarction, ischaemic or haemorrhagic stroke, or peripheral arterial ischaemia, among others), kidney failure, chronic liver disease or HIV/AIDS.
⇒ Active endocrinological disease, innate errors of metabolism, myopathies or epilepsy.
⇒ Patients who have undergone bariatric surgery surgical techniques or used for the treatment of other pathologies (eg, 'Roux Y').
⇒ Rheumatoid arthritis, Parkinson's disease, active cancer treatment in the past year or another medical condition in which fasting is contraindicated.
⇒ Use of medications that may affect the results of the study such as drugs for glycaemic control (eg, antidiabetic, steroids, beta-blockers, antibiotics, prebiotics, probiotics and symbiotics).
⇒ Diagnosis of major sleep or eating disorders.
⇒ Caregiver for a dependent requiring frequent nocturnal care/sleep interruption or shift workers with nocturnal hours.
⇒ Metal or electrical prosthesis.
⇒ Foreign bodies in the eyes.
⇒ Fear of needles and claustrophobia to MRI.
⇒ Active tobacco or illicit drug use or a history of alcohol abuse treatment (ie, moderate or severe alcoholism).
⇒ Participating in a weight loss or a supervised exercise programme (ie, >30 min in three times per week, or >45 min in two or more times per week at moderate-to-vigorous intensity).
⇒ Pregnancy and lactation or planned pregnancy (within the study period).
⇒ Frequent travel over time zones during the study period.
⇒ Being unable to understand and to accept the instructions or the study objectives and protocol.
⇒ Not having or being able to use a smartphone with Apple iOS or Android OS.
⇒ Are deemed unsuitable by the investigator for any other reason.

effective in achieving a balanced sex distribution.[23–25]

– Sample size calculations: Our sample size calculations have taken into account subgroup analyses by sex. We have conservatively estimated a maximum drop-out rate of approximately 20%. By considering this drop-out rate, we have ensured that our study is adequately powered to detect the specified effect size even if there are differential drop-out rates between men and women. For example, if men have a drop-out rate of 5% and women have a drop-out rate of 15%, our study will still have sufficient power to analyse the data separately in men and women.

– Expected drop-out rate: While we have conservatively estimated the maximum dropout rate, we anticipate that the actual drop-out rate will be relatively lower and similar between both sexes. This expectation is based on our previous studies and the measures we have to promote participant engagement and adherence.[23–25]

### Randomisation and blinding

We expect to recruit and randomise ~184 participants using both stratification and permuted blocks with random block sizes, after completing the preintervention measurements. Randomisation will be stratified based on sex (men–women), resulting in two strata. Prior to the beginning of the trial, randomisation lists will be generated for each stratum. The block sizes will be randomly determined, with allowable sizes of 4 and 8. Within each block, random selection will be used to assign participants to one of the four possible groups (ie, UC, TRE, Exercise or TRE+Exercise) using a parallel design with a 1:1:1:1 allocation ratio. The sequential assignment of participants will follow the predetermined randomisation list specific to their strata. The utilisation of random block sizes ensures that the next assignment cannot be predicted. Stratification by sex ensures that the intervention groups will be balanced for this important characteristic. Blinding procedures will be rigorously applied to all personnel responsible for assessing primary outcomes, specifically hepatic fat depots and other ectopic fat depots derived from MRI. This blinding will also extend to those analysing cardiometabolic risk factors, conducting glucose monitoring, examining faecal microbiota, evaluating physical activity and sedentary time, as well as those engaged in the statistical analysis of data. Conversely, personnel responsible for other measurements and intervention administration will not be blinded to the group assignment (open label). Participants will receive clear information about the group they will be assigned to, along with details about the study hypotheses. In order to ensure practicality and feasibility, and based on previous research experience,[23 24 26] the study will be conducted in successive waves of participants, each of them including a maximum of 16 participants.

### Intervention description
#### UC group

Participants randomly assigned to the UC group will be indicated to continue with their dietary eating time schedule. All participants will receive monthly in-person nutritional education sessions, lasting approximately 90 min, throughout the intervention for weight management and cardiovascular health promotion based on Mediterranean dietary patterns[27] and physical activity recommendations from the WHO.[28] Key points in nutritional counselling will be: (a) to correctly interpret food labels and to plan grocery, (b) to recognise and include high satiety foods and (c) to fight against some extended myths in Spanish culture such as 'alcohol in small quantities is healthy', 'it is unhealthy to skip breakfast' or 'it is important to eat five meals per day'.

## TRE intervention

Before the beginning of the intervention, participants allocated to TRE groups will select their preferred 8-hour eating window before the intervention and will be required to maintain the same 8-hour eating window during the 12-week intervention. They will be advised that the last meal should be completed before or at 21:00. Participants assigned to the TRE groups will be strictly prohibited from consuming any calorie-containing food or beverage outside their designated 8-hour eating window. However, they will be allowed to consume water, coffee and tea without sugar or artificial sweeteners during the fasting period. Instructions for the TRE intervention will emphasise its daily implementation, meaning participants are expected to adhere to the eating window restriction (±30 min) throughout all 7 days of the week.

## Supervised exercise intervention

The objective of the supervised exercise programme will be to ensure its transferability and feasibility for the target population. We will follow the Consensus on Exercise Reporting Template to facilitate replicability and transparency[29] and record the timing when participants are training.[30] Thus, the physical activity recommendations for adults proposed by the WHO serve as the foundation for determining the specific exercise dosage in TEMPUS.[28] Given that both resistance and aerobic training modalities have shown improvements in hepatic steatosis among patients with MASLD,[31 32] the TEMPUS study will combine both supervised resistance and aerobic high-intensity interval training (HIIT) sessions. Qualified sports scientist from outside the research group will carefully supervise the exercise sessions and work with groups of no more than six persons to ensure that participants perform the exercise technique correctly and at a proper intensity. Moreover, participants will receive an individualised moderate-intensity goal-setting aerobic (ie, walking) programme consisting of increasing 10% daily steps per week based on their daily steps logs. Goal-setting aerobic will be updated weekly using a smart band (Xiaomi Mi Band 7, Xiaomi, Pekin, China) that participants enrolled in the exercise groups will wear on their wrists during the intervention period.

### Volume

Participants will engage in a 12-week intervention programme including two supervised exercise sessions per week ranging from 60 to 90 min per session. Furthermore, participants will receive instructions to complete their personalised daily step goals established for each day.

### Intensity

Supervised sessions will start with a supervised circuit-based resistance training that focuses on upper and lower body exercises that target major muscle groups. To prescribe intensity, the rate of perceived exertion (RPE) scale ranging from 0 to 10 will be used, using the

evaluations and first sessions to properly train the participants on this scale.[33] The target intensity for resistance training will be set at an RPE level greater than 7, which will vary depending on the week number of the intervention (from 7 to 10). The participants will be encouraged to achieve this intensity through all the exercises performed in each lap of the circuit. Moreover, training sessions will include HIIT as the vigorous-intensity aerobic exercise component. HIIT will consist of 3–4 sets of 4 min intervals at >85% of the individual heart rate reserve (HRR) with 4 min of active recovery at 65%–75% HRR. HRR will be calculated considering the peak of HR achieved in the incremental treadmill test and the rest HR lying in bed recorded during preintervention assessments. HR will be continuously monitored during all exercise sessions using the Polar H10 band (Polar Products, Stow, Ohio, USA).

### Frequency

Participants will perform two supervised exercise training sessions per week, with a resting period of at least 48 hours between them. Additionally, participants will be instructed to accomplish the prescribed daily step goal. If a participant misses a training session, it will be rescheduled and recovered considering a minimum resting period of 48 hours between sessions.

### Type of exercise

The resistance training sessions will be composed of major upper and lower body muscle groups including[34]: horizontal/vertical pull exercises (ie, seated low row and lat pulldown with resistance elastic bands), hip-dominant exercises (ie, deadlift with resistance elastic bands and weight-bearing glute bridge), knee-dominant exercises (ie, weight-bearing squats and lunges) and horizontal/vertical push exercises (ie, weight-bearing push-ups and shoulder press with resistance elastic bands). We will propose four levels of exercises' difficulty based on the resistance elastic band and the progressive complexity of the basic movement patterns involved (online supplemental figure 1). After the resistance circuit training, HIIT will be preferably performed on a treadmill to ensure a suitable progressive overload. As an alternative modality, elliptical ergometer will be used.

### Training load variation

We are aware that participants might not be immediately capable of exercising at high intensities and volumes. Therefore, there will be a gradual progression to the assigned exercise dose in three phases (see figure 2) and a proper technique will be a priority for the difficulty level progression to avoid potential injuries.

### Training periodisation

The supervised exercise intervention will be divided into three phases (figure 2). The initial phase will have a length of 6 weeks and will start with a first week of familiarisation. During this week, participants will learn the structure and organisation of the sessions and the movement patterns that constitute the base of the different

| Phase | | FIRST | | | | | | SECOND | | | | THIRD | |
|---|---|---|---|---|---|---|---|---|---|---|---|---|---|
| Week | | 1* | 2 | 3 | 4 | 5 | 6 | 7 | 8 | 9 | 10 | 11 | 12 |
| Supervised Exercise Volume (min/week) | | 110 | 130 | 130 | 130 | 130 | 130 | 150 | 150 | 150 | 150 | 170 | 170 |
| **Warm-up (RT)** | Volume (min/session) | 5 | 5 | 5 | 5 | 5 | 5 | 5 | 5 | 5 | 5 | 5 | 5 |
| **Resistance training** | Total volume (min/week) | 32 | 32 | 32 | 32 | 32 | 32 | 48 | 48 | 48 | 48 | 64 | 64 |
| | Intensity (RPE 1-10 scale) | 7 | 7 - 8 | 7 - 8 | 7 - 8 | 8 - 9 | 8 - 9 | 8 - 9 | 8 - 9 | 9 - 10 | 9 - 10 | 9 - 10 | 9 - 10 |
| | Secs work/Secs rest (per exercise) | 30/30 | 30/30 | 30/30 | 30/30 | 30/30 | 30/30 | 30/30 | 30/30 | 30/30 | 30/30 | 30/30 | 30/30 |
| | Number of laps (per session) | 2 | 2 | 2 | 2 | 2 | 2 | 3 | 3 | 3 | 3 | 4 | 4 |
| | Rest between laps (min) | 2 | 2 | 2 | 2 | 2 | 2 | 2 | 2 | 2 | 2 | 2 | 2 |
| | Number of exercises (per session) | 8 | 8 | 8 | 8 | 8 | 8 | 8 | 8 | 8 | 8 | 8 | 8 |
| **Rest between RT & AET** | Volume (min/session) | 3 | 5 | 5 | 5 | 5 | 5 | 5 | 5 | 5 | 5 | 5 | 5 |
| **Warm-up (HIIT)** | Volume (min/session) | 2 | 2 | 2 | 2 | 2 | 2 | 2 | 2 | 2 | 2 | 2 | 2 |
| **HIIT** | Vigorous-intensity intervals / Moderate-intensity intervals | 3/3 | 4/4 | 4/4 | 4/4 | 4/4 | 4/4 | 4/4 | 4/4 | 4/4 | 4/4 | 4/4 | 4/4 |
| | Total volume vigorous-intensity (min/week) | 24 | 32 | 32 | 32 | 32 | 32 | 32 | 32 | 32 | 32 | 32 | 32 |
| | Intensity (%HRR) | 80 - 85 | 85 - 95 | 85 - 95 | 85 - 95 | 85 - 95 | 85 - 95 | 85 - 95 | 85 - 95 | 85 - 95 | 85 - 95 | 85 - 95 | 85 - 95 |
| | Total volume moderate-intensity (min/week) | 24 | 32 | 32 | 32 | 32 | 32 | 32 | 32 | 32 | 32 | 32 | 32 |
| | Intensity (%HRR) | 60 - 65 | 65 - 75 | 65 - 75 | 65 - 75 | 65 - 75 | 65 - 75 | 65 - 75 | 65 - 75 | 65 - 75 | 65 - 75 | 65 - 75 | 65 - 75 |
| **Cool down** | Volume (min/session) | 5 | 5 | 5 | 5 | 5 | 5 | 5 | 5 | 5 | 5 | 5 | 5 |

**Figure 2** Supervised exercise periodisation of the TEMPUS project. *Week of familiarisation. HIIT, high-intensity interval training; HRR, heart rate reserve; RPE, rate of perceived exertion; RT, resistance training. AET, aerobic endurance training.

exercises. Considering that an inactive person may be unable to immediately train at the selected doses, the familiarisation phase will prepare participants to gradually increase the workload until the required dose will be achieved. On the other hand, the resistance training will set light loads using resistance elastic bands or weight-bearing exercises. They will increase the load and coordinative difficulty (through four levels for each exercise) as soon as participants perform the exercises with a proper technique.

We anticipate that both resistance training and HIIT load will be parallelly higher with the participants fitness increase. In this context, HIIT will be controlled based on HRR needing superior speed or power has to be selected to achieve a determined percentage of HRR when fitness levels are increasing.

We expect a concurrent increase in the intensity of both resistance training and HIIT as participants' fitness levels improve. Specifically, adjustments (ie, greater speed or power) to HIIT will be meticulously calibrated based on the HRR. This approach is essential to maintain a targeted percentage of HRR, particularly as participants exhibit enhanced fitness capacities.

*Training session*

A training session will be organised as follows (see example in online supplemental figure 2): (1) a warm-up of one circuit training lap with light loads using resistance elastic bands or weight-bearing exercises (20 s work: 20 s rest),

(2) 2–4 circuit training laps of eight exercises (30 s work: 30 s rest) at 7–10 RPE, with a between-set rest of 3–5 min and (3) 3 or 4 sets of 4 min intervals at 85%–95% HRR interposed by 4 min of active recovery at 65%–75% of HRR, with 2 min prior of warm-up. A cool-down protocol of 5 min will be performed at the end including anterior and/or posterior chain exercises for muscle elongation and/or relaxation. The total duration of exercise sessions will be 55 min in the familiarisation week, 65 min during the first phase, 75 min in the second phase and 85 min in the third phase.

**Outcome measures**

A 2-week lead-in period will be implemented prior to preintervention assessments and group allocation (see figure 1). During this period, potential participants will be instructed to maintain their usual dietary and physical activity habits. They will be provided with a mobile phone application (Tempus: com.nnbi.app_extreme_granada; NNBi2020 S.L., Navarra, Spain) to record daily information regarding their eating time, sleeping patterns, naps and any adverse events experienced. Additionally, physical activity and sleep quality will be evaluated using accelerometry, and glucose excursions over day and night will be assessed using continuous glucose monitoring (CGM) devices over the course of 2 weeks. This data collection will be used to verify that the participants' habitual eating windows are ≥11 hours and perform >30 min in three times per week, or >45 min in two or more times

**Table 1** Overview of study outcomes

| Outcomes | Preintervention | Mid-term | Postintervention | 12-month follow-up |
|---|---|---|---|---|
| Primary outcome | | | | |
| Hepatic fat | ✓ | | ✓ | ✓ |
| Secondary outcomes | ✓ | | | |
| Ectopic fat depots | ✓ | | ✓ | ✓ |
| Anthropometry and body composition | ✓ | ✓ | ✓ | ✓ |
| Cardiometabolic markers | ✓ | | ✓ | ✓ |
| Glycaemic control | ✓ | | ✓* | |
| Faecal microbiota | ✓ | | ✓* | |
| Plasma and faecal metabolomics | ✓ | | ✓ | ✓ |
| Sleep and physical activity | ✓ | | ✓* | |
| Psychosocial | ✓ | | ✓ | |
| Quality of life | ✓ | | ✓ | ✓ |
| Eating behaviour and dietary habits | ✓ | | ✓ | ✓ |
| Fitness | ✓ | | ✓ | |

*Assessment will be conducted during the last 2 weeks of the intervention.

per week at moderate-vigorous intensity activity (see inclusion and exclusion criteria, box 1). Preintervention measurements, after 12-week intervention measurements (±3 days) and after 12 months postintervention measurements will be conducted by a dedicated team of trained staff to ensure consistency and reliability. In addition, we will assess anthropometry and body composition at week 6 (mid-term in figure 1). By including these intermediate measurements, we will potentially capture changes over time and evaluate the trajectory of outcomes during the 12-week intervention period. A detailed overview of the study's outcomes can be found in table 1, which provides a comprehensive summary of the variables assessed.

The study primary outcome will be changes from baseline to 12 weeks in hepatic fat. Secondary outcome variables include changes from baseline to 12 weeks in other ectopic fat depots (ie, abdominal visceral and SAT, pancreatic fat and mid-thigh intermuscular and intramuscular adipose tissue), anthropometry and body composition (ie, fat mass and fat free mass, bone mineral density and content), cardiometabolic markers (ie, glucose, insulin, haemoglobin A1c, lipid profile, vitamin D, alkaline phosphatase and calcium, liver and kidney function markers, estimated glomerular filtration rate and steroid and thyroid hormones), 24 hours, diurnal and nocturnal mean glucose (via CGM), faecal microbiota, sleep and physical activity patterns, psychological outcomes (ie, depression, stress and anxiety), quality of life, reproductive profile and status, dietary habits and eating behaviour assessment, and fitness (ie, cardiorespiratory fitness and muscular strength). After the screening examination and the 2-week lead-in period, preintervention assessments will be held within three face-to-face sessions:

Day 1: Participants will perform physical fitness tests and will be required to complete several web-based questionnaires. These questionnaires will gather information on several outcomes such as quality of life, sleeping patterns, psychological status and eating behaviour. Dietary intake will also be assessed by qualified nutritionist where participants will be asked to provide the first 24-hour recall detailing their food consumption.

Day 2: Fasting venous blood samples will be collected at the San Cecilio University Hospital (Granada, Spain). Thereafter, anthropometric and body composition measurements will be conducted at the Sport and Health University Research Institute (iMUDS—300 m apart). Finally, oral glucose tolerance tests will be also conducted as part of the glycaemic assessment.

Day 3: MRI and elastography will be performed in the San Cecilio University Hospital by the TEMPUS' medical staff (Granada, Spain). To ensure consistency and standardisation during the assessment, participants will be instructed to maintain a fasting period of 8–10 hours prior to the appointment, during which they will be advised to consume only water and abstain from solid food or other beverages. In addition, participants will be advised not to consume alcohol or diuretics 24 hours before the test and to avoid stimulants like caffeine for 12 hours before the test.

### Hepatic fat (primary outcome)
The quantification of hepatic fat and iron content will be performed using MRI with a Siemens 3T Magnetom Vida scanner located at San Cecilio University Hospital in Granada. Additionally, liver steatosis, viscosity and fibrosis severity will be assessed using attenuation imaging, shear

wave elastography and shear wave dispersion with a Aixplorer mach 30, Hologic. The Fibrosis-4 index will be calculated as an indicator of liver fibrosis severity.[35]

### Ectopic fat depots

Abdominal subcutaneous, visceral and intermuscular adipose tissue, as well as pancreatic fat content, will be obtained using MRI. To assess subcutaneous, visceral and intermuscular adipose tissue in all 3D abdominal volume (ie, volume, cross-sectional area at selected levels and mean/median fat fraction), a semiautomatic software program will be employed for tissue segmentation. These valuable image markers will be derived from a standard 6 echo Dixon series, ensuring accurate characterisation of abdominal adipose tissue distribution and composition. We will also measure image markers from mid-thigh using a 6 echoes Dixon series and will obtain cross-sectional area, muscular tissue, intramuscular adipose tissue, intermuscular adipose tissue, fat fraction, SAT and bone marrow fat fraction. The segmentation for all these structures will be performed with a semiautomatic proprietary algorithm.

### Anthropometry and body composition

The high prevalence of MASLD among adults with obesity adults justify the need of analysing anthropometric and body composition variables. Body weight and height measurements will be obtained using a stadiometer and scale (Seca model 799, Electronic Column Scale, Hamburg, Germany). Participants will be instructed to be barefoot and wear lightweight clothing during these measurements. Neck, waist and hip circumferences, as well as calf girth, will be determined following the procedures outlined by the International Society for the Advancement of Kinanthropometry (ISAK) by certified personnel (ID's ISAK: #637686045477240182 and #638227824311214767).[36] Furthermore, bone mineral density, fat mass, fat free mass and overall adipose mass will be assessed by dual-energy X-ray absorptiometry scans (Discovery Wi, Hologic, Bedford, Massachusetts, USA) and phase angle will be assessed by bioelectrical bioimpedance (Tanita MC 980-MA Plus, Tanita, Tokyo, Japan). Participants will be instructed to maintain a fasting period of 8–10 hours prior to the appointment, during which they will be advised to consume only water and abstain from solid food or other beverages. In addition, participants will be advised not to consume alcohol or diuretics 24 hours before the test and to avoid stimulants like caffeine for 12 hours before the test, and avoid engaging in moderate exercise or physical activity for 24 hours, or vigorous exercise for 48 hours, prior to the test.

### Cardiometabolic risk markers

Venous blood samples will be carefully preserved at a temperature of −80°C to ensure their integrity for subsequent analysis. We will collect a full set of cardiometabolic risk markers, as these parameters may inform about the metabolic dysregulation in liver disease.[37] These set will include: fasting glucose (Alinity C system analyzer, Abbott Laboratories, Illinois, USA), insulin (UniCel DxI 800 access immunoassay system, Beckman Coulter, California, USA), haemoglobin A1c (automated glycohaemoglobin G11 analyser, Horiba) and lipid profile (ie, total cholesterol, high-density lipoprotein cholesterol, low-density lipoprotein cholesterol, triglycerides, apolipoprotein A1 and B) using Alinity C system analyzer (Abbott Laboratories). Low-density lipoprotein cholesterol will be calculated using a previously validated equation (LDL-c=CT−HDL-c−(TG/5). Furthermore, we will measure vitamin D, alkaline phosphatase and calcium (Alinity C system analyzer, Abbott Laboratories). Liver and kidney function markers (ie, alanine transaminase, gamma-glutamyl transferase, bilirubin, creatinine) will be also measured using an Alinity C system analyzer, while estimated glomerular filtration rate, steroid hormones (ie, oestradiol, progesterone, testosterone, follicle stimulating hormone and luteinising) and thyroid hormones (ie, thyrotropin, thyroxine, triiodothyronine) will be also assessed using a UniCel DxI 800 access immunoassay system (Beckman Coulter). Blood count and biochemistry and inflammatory markers (ie, iron, ferritin, folic acid, C reactive protein and interleukin-6) will be measured using a AU5800 automated analyzer (Beckman Coulter). Additionally, we will calculate insulin resistance surrogates, such as the homeostatic model of assessment for insulin resistance and the quantitative insulin-sensitivity check index. Finally, we will also conduct untargeted metabolomics analysis on plasma samples. The omics analyses will allow the identification of: (1) metabolites strongly associated to hepatic fat content (baseline—cross-sectional analysis) and (2) metabolites predictors of hepatic fat content changes after the intervention. In addition, systolic and diastolic blood pressure will be measured using an automated monitor (M3-Comfort, Omron Healthcare Europe B.V., Hoofddorp, The Netherlands), following the established guidelines outlined by the 2021 European Society of Hypertension.[38]

### Glycaemic control

Due to the crucial role of insulin resistance in MASLD development, through fatty acid accumulation in hepatocytes, CGM and analysis of acute response to glucose will be measured. Participants will be instructed to wear a CGM device (FreeStyle LibrePro, Abbott Laboratories, Abbott Park, IL) during 2 weeks before the intervention (lead-in period) and during the last 2 weeks of the intervention (weeks 11 and 12). The CGM data obtained will be used to calculate different variables related to glycaemic control (ie, 24-hour mean glucose), following the guidelines outlined in the most recent international consensus statement.[39] Additionally, oral glucose tolerance tests will also be conducted as part of the glycaemic assessment using a 75 g oral glucose dose (NUTER TEC: orange flavour, Toulouse, France).

## Faecal microbiota

For the comprehensive identification and quantification of faecal microbiome diversity and composition, preintervention and during the last 2 weeks of the intervention, faecal samples will be collected. Stool microbial DNA will be isolated from participants and subsequently, 16S rRNA gene amplicon sequencing, with the possibility of employing shotgun methodology pending final budget considerations, will be performed. Furthermore, a faecal metabolomic fingerprint analysis will be conducted to determine the metabolic profile among different groups of patients.

## Sleep and physical activity

Sleep quality and chronotype will be evaluated through the administration of validated questionnaires, including the Pittsburgh Sleep Quality Index,[40] Munich Chronotype Questionnaire[41] and Horne and Östberg Questionnaire for Morning-Evening type assessment.[42] Objective measures of sleep and physical activity levels will be obtained using triaxial accelerometer (ActiGraph GT3X+, Pensacola, Florida, USA) that participants will wear on their non-dominant wrist during 2-week periods: before the beginning of the intervention (lead-in period) and during the last 2 weeks of the intervention (weeks 11 and 12).

## Psychosocial assessment

As weight loss intervention may improve emotional well-being and psychosocial functioning, and therefore, several key psychological dimensions will be evaluated through the administration of validated questionnaires: the Beck Depression Inventory Fast Screen for depression,[43] the Perceived Stress Scale for stress (PSS)[44] and the State-Trait Anxiety Inventory for anxiety.[45]

## Quality of life

Quality of life may also improve from weight loss intervention[46] and will be evaluated using the EuroQol Five-Dimension Five-Level questionnaire[47] and Rand Short Form-36.[48] In addition, participants will be asked to complete an adverse events questionnaire to identify any potential adverse effects or complications experienced during the intervention.

## Eating behaviour and dietary habits

To assess participants' adherence to the Mediterranean dietary pattern, validated questionnaires such as the PREDIMED questionnaire will be administered. This questionnaire provides a reliable measure of adherence to the specific dietary components and guidelines of the Mediterranean diet.[49] Food Frequency Questionnaire[50] will be administered to assess the frequency that participants have consumed each specific food during the previous 4 weeks. Food Craving Inventory[51] and the Adult Eating Behaviour Questionnaire for appetite traits[52] will be also administered.

## Fitness

An incremental treadmill exercise test until exhaustion will be performed to determine cardiorespiratory fitness. The modified Balke protocol[53] will be applied, which has been extensively used and validated.[54–56] The exercise ECG and HR will be monitored continuously and reviewed by a cardiologist. Furthermore, capillary blood lactate will be measured at rest and along different stages of the test (Lactate Pro 2 LT-1730, Arkray, Kyoto, Japan). Supervised exercise proposed in this study could restore or mitigate adverse effect of diet-induced weight loss on muscle strength.[57] Thus, upper muscular strength will be assessed through hand grip strength test[58] using a digital hand dynamometer (TKK 5401 Grip-D; Takei, Tokyo, Japan), whereas lower body muscular strength will be assessed through the 30 s sit-to-stand muscle power test[59] and walking speed to assess functional capacity with gait speed test.[60]

## Confounding

As unintentional reductions in energy intake (10%–30% or ~300–500 kcal/day) have been reported when participants confine their eating windows to 4–10 hours/day,[61 62] we will control and analyse changes in energy intake over the intervention period. Participants will undergo dietary assessments through the completion of 3 nonconsecutive 24-hour dietary recalls (two working days and one non-working day).[63] These recalls will be conducted via face-to-face or telephone interviews by qualified and trained research nutritionists. Total energy intake, carbohydrates, fat and protein intake will be calculated. In addition, to understand the important sex dimorphism in MASLD development, at baseline and after the intervention, specific reproductive-profile questions will be asked regarding detailed information on menstrual cycle history and hormonal contraceptive use and type, as well as regarding any gynaecological condition. This will help in categorising participants into different hormonal profiles, following the recent consensus.[64–66]

## Participant retention, adherence and sustainability

The principal investigator and study team will exert every effort to facilitate participants' completion of all study visits and ensure overall study retention. The following strategies will be implemented to maximise participants' retention and minimise loss to follow-up: (a) implementing a proactive retention plan that focuses on building close participant relations and ensuring participant satisfaction, (b) giving the opportunities for participants and their families to ask questions and voice any concerns related to their condition throughout the study, (c) reinforcing comprehension of the objectives and protocol of the study during study visits or conducting question and answer sessions after each visit and (d) evaluating each likelihood of drop-out and implementing appropriate interventions to maintain their interest and motivation to continue participating in the study.

All supervised exercise sessions will be performed in a well-lit and airy room, providing to the participants the opportunity to choose their own music. The training specialists and other study staff will consistently offer support to participants throughout the duration of the study.

During the 12-week intervention period, participants will be required to record their daily sleep and eating times (ie, exact times of the beginning of the first meal and of the end of the last meal), as well as any potential adverse events in a mobile phone app specifically designed for the study. These data will be revised 2–3 times every week, asking the participant for missing records, and will provide insights into their adherence to the prescribed eating window. Participants in the exercise groups and TRE groups will be labelled as 'adherent' if they perform >80% of training sessions and eat within their allocated window of 8 hour (±30 min) >80% of days. Finally, we will assess the long-term adherence to the intervention (at the 12-month follow-up). This will allow us to evaluate the sustainability of participants' adherence over an extended period.

### Adverse events

The study coordinators and project managers will oversee the collection of data and monitor the frequency of reported adverse events on a weekly basis using the mobile phone app (Tempus: com.nnbi.app_extreme_granada; NNBi2020 S.L., Navarra, Spain). Additionally, participants will complete validated questionnaires that assess gastrointestinal and autonomic symptoms, well-being, eating behaviour, sleep quality, stress levels, mood, anxiety and depression. These questionnaires will offer valuable information on any potential adverse effects and overall health-related outcomes. If a serious adverse event or an unanticipated problem occurs, the study coordinators will immediately notify both the principal investigators and the medical staff. Subsequently, a collective decision will be made and, if needed, the ethics committee will be properly informed. Moreover, appropriate measures will be taken to address and manage the reported event effectively.

### Analytical approach and data management

The effects on primary and secondary outcomes in response to the present 12-week intervention will be assessed based on repeated-measures linear mixed-effects multilevel models.[67] Individual measures of change will be, therefore, modelled as the function of the randomly assigned group, assessment time and their interaction terms. All the analyses will be conducted separately for men and women. Model-based estimations will be performed with an intention-to-treat approach (primary analyses) using the restricted maximum-likelihood method, the model assuming that missing values are missing-at-random. Analyses and estimations will also be performed with a per-protocol approach and an attrition propensity will be calculated using a logistic model

predicting attrition with baseline values of allocation group, age, sex and BMI. Additional models will be conducted including energy intake, physical activity or reproductive status in women. In addition to the conventional approach of assessing intervention effects based on statistical and practical significance, it is important to highlight that this study will employ a practical benefit approach. This approach emphasises the reporting of unadjusted values that are intuitive to human judgement and easily replicable, considering the design and methodology of the study.

Data collected will be directly entered into REDCap (Research Electronic Data Capture), a secure web-based platform specifically designed to create and manage research-related databases and surveys. This platform will ensure data security and confidentiality. For any data not recorded in REDCap, strict access control measures will be implemented to securely store the data on university computers, maintaining confidentiality and data integrity.

To ensure data quality and integrity, regular quality control checks will be conducted to identify any potential data anomalies, such as missing data or forms, data that falls outside the expected range, erroneous data entries, illogical dates over time, data inconsistencies across different forms and study visits and incomplete fields on completed forms without a valid explanation for the missing data. Any identified issues will be promptly reviewed and addressed by the data monitoring committee to ensure the accuracy and reliability of the data. The data monitoring committee will be independent from the sponsor. There are no auditing procedures planned.

### Patient and public involvement

TEMPUS will stimulate patient and public involvement throughout the entire process. Potential participants have been involved in the preparation of the intervention and in the development the mobile app, organisation of the outcome measures, as well as are helping in the recruitment of new participants. Once we have the results ready, we will actively involve the participants in the reporting and advocacy of the study results.

## ETHICS AND DISSEMINATION

The study has received ethical approval from the Granada Provincial Research Ethics Committee (CEI Granada— 0365-N-23) and will be performed following the ethical guidelines of the Declaration of Helsinki. Before their inclusion in the study, participants will be required to provide oral and written informed consent, with further details available in Online supplemental patient consent form.

Results will be presented in peer-reviewed, scientific journals and at international conferences. We aim to publish a main paper with the primary outcome data. Since it is a large study with numerous secondary

outcomes, other specific manuscripts on each topic will also be submitted for publication.

## DISCUSSION

Data from National Spanish registries indicate that the overweight/obesity epidemic is reaching rates of 70% among men and 50% among women[68] and that nearly 90% of adults with overweight/obesity have MASLD.[69] The TEMPUS study is clinically and socially urgent because E.U. member states annually spend around €80.4 billion to treat diseases caused by the lack of physical activity and unhealthy nutrition,[70] directly linked to obesity.

This study will provide strong scientific evidence to overcome the shortcomings found in this field. A major limitation of current strategies for preventing and treating obesity is the short length of interventions and low adherence rates and, thus, the poor medium and/or long-term efficacy.[16 17 19 71] TEMPUS will be the first study to quantify TRE's long-term effects on hepatic fat and cardiometabolic health separately in men and women. Indeed, acknowledging and thoroughly examining the existing sex-related differences will contribute to a deeper comprehension of the variations in obesity and associated comorbidities between the sexes. This understanding will play a pivotal role in developing tailored, sex-specific treatments and therapies, improving therefore the effectiveness and precision of interventions for individuals. In addition, TEMPUS will offer practical and effective solutions, easy for clinicians to deliver and intuitive for patients to implement and maintain throughout their lives. Regarding sustainability, TRE presents a practical advantage compared with existing stringent energy-restricted diet interventions, as it eliminates the need for specific calorie restrictions or discretionary food choices. Nonetheless, participants will be advised to follow the Mediterranean dietary pattern and adhere to physical activity recommendations for weight loss and overall health promotion. In turn, this will help reducing or even preventing undesirable adverse effects linked to caloric restriction, such as hunger and weight regain. Cutting-edge technologies—including metagenomics and metabolomics—will allow us to discover the faecal microbiota composition and functionality fingerprint associated with TRE and supervised exercise-induced beneficial effects. This will give us the possibility to elucidate potential mechanisms behind the potential health-related improvements derived from TRE+Exercise in humans' health.[54]

TEMPUS will contribute to several WHO Sustainable Development Goals[72] including the promotion of healthy lives and well-being, gender equality, partnership for sustainable development, through the multidisciplinary collaboration to achieve goals, and sustainable and safe cities. Through the implementation of exercise interventions, it will also raise awareness of the importance of active transportation. TEMPUS will potentially contribute to establish an easy-to-implement and pragmatic non-pharmacological intervention to effectively reduce hepatic fat and, thus, improve cardiometabolic health of adults with obesity. Furthermore, our study is expected to unveil the mechanisms by which TRE combined with supervised exercise may result in the aforementioned outcomes.

The TEMPUS has several strengths, including the possibility to determine whether TRE combined with exercise is superior to TRE or exercise alone on reducing hepatic fat and improving cardiometabolic health in adults with obesity. This study is powered and designed to evaluate the effect of the interventions separately in men and women. Moreover, TEMPUS will quantify the persistence of the effects of a 12-week TRE and exercise intervention on hepatic fat and cardiometabolic health in the long term, with a follow-up at 12 months. However, the study is limited to adults with obesity aged 25–65 years, and its results cannot be extended to older populations or to people with other pathologies. Future studies should also increase the length of the intervention beyond 3 months.

**Author affiliations**
[1]Department of Physical Education and Sports, Faculty of Sport Sciences, Sport and Health University Research Institute (iMUDS), University of Granada, Granada, Spain
[2]Department of Physiology, Faculty of Pharmacy, Institute of Nutrition and Food Technology, Biomedical Research Centre, University of Granada, Granada, Spain
[3]Instituto de Investigación Biosanitaria, Ibs, University of Granada, Granada, Spain
[4]Department of Physiology, Faculty of Medicine, University of Granada, Granada, Spain
[5]Centro de Investigación Biomédica en Red Fisiopatología de la Obesidad y Nutrición (CIBERobn), Instituto de Salud Carlos III, Madrid, Spain
[6]LFE Research Group, Department of Health and Human Performance, Faculty of Physical Activity and Sport Science (INEF), Universidad Politécnica de Madrid, Madrid, Spain
[7]Servicio de Análisis Clínicos, Hospital Universitario San Cecilio, Granada, Spain
[8]Servicio de Radiodiagnóstico, Hospital Universitario San Cecilio, Granada, Spain
[9]Department of Psychology, Universidad Loyola Andalucía, Sevilla, Spain
[10]Servicio de Microbiología, Hospital Universitario San Cecilio, Granada, Spain
[11]Centro de Investigación Biomédica en Red de Enfermedades Infecciosas (CIBERinfecc), Instituto de Salud Carlos III, Madrid, Spain
[12]Centro de Investigación Biomédica en Red de Enfermedades Hepáticas y Digestivas (CIBERehd), Instituto de Salud Carlos III, Madrid, Spain
[13]Department of Pharmacology, School of Pharmacy, University of Granada, Granada, Spain
[14]Department of Pharmacology, Center for Biomedical Research, Granada, Spain
[15]Department of Electrical, Electronic and Communications Engineering, Public University of Navarre, Pamplona, Spain
[16]Endocrinology and Nutrition Unit, Hospital Universitario Virgen de las Nieves, Granada, Spain
[17]Endocrinology and Nutrition Unit, Hospital Universitario San Cecilio, Granada, Spain
[18]Department of Medicine, Faculty of Medicine, University of Granada, Granada, Spain
[19]Centro de Investigación Biomédica en Red Fragilidad y Envejecimiento Saludable (CIBERfes), Instituto de Salud Carlos III, Madrid, Spain
[20]Navarre Institute of Health Research, Pamplona, Spain
[21]Institute for Sustainability & Food Chain Innovation, Department of Health Sciences, Public University of Navarre, Pamplona, Spain

**Contributors** AC-C, AC-J and JRR were responsible wrote the first draft of the article. JJM-O, FA-G, RCC, MTGC, PVGP, CH-M, RS-L, AD-I-O, AL-V, MM-F, AC-B, FG, AR-N, JJGP, RCC, JLM-R, AM-G, MMT and IL contributed to the study design and revised the protocol critically for important intellectual content. AC-C, AC-J and JRR coordinated the finalisation of the draft by integrating the inputs of all authors. All

authors have read and approved the final version of this manuscript and agreed to be accountable for all aspects of the work.

**Funding** This study is funded by the Spanish Ministry of Science, Innovation and Universities (PID2022-141506OB-I00) and the European Regional Development Funds (ERDF), Agencia Estatal de Investigación; the University of Granada Plan Propio de Investigación-Excellence actions: Unit of Excellence on Exercise Nutrition and Health (UCEENS). AC-C is supported by the Spanish Ministry of Science and Innovation (FJC2020-043385-I). AC-J is supported by the Spanish Ministry of Universities (FPU21/01161). JJM-O is supported by the Spanish Ministry of Universities (FPU22/01631)RC is supported by a grant for the Requalification of the Spanish University System 2021–2023 from the Spanish Ministry of Universities (RD 289/2021), funded by the European Union-Next Generation EU. This study is part of a PhD Thesis conducted in the Biomedicine Doctoral Studies of the University of Granada.

**Competing interests** None declared.

**Patient and public involvement** Patients and/or the public were involved in the design, or conduct, or reporting, or dissemination plans of this research. Refer to the Methods section for further details.

**Patient consent for publication** Consent obtained directly from patient(s).

**Provenance and peer review** Not commissioned; externally peer reviewed.

**ORCID iDs**
Alba Camacho-Cardenosa http://orcid.org/0000-0002-7682-8336
Antonio Clavero-Jimeno http://orcid.org/0000-0002-6135-9848
Juan J Martin-Olmedo http://orcid.org/0009-0005-8072-0116
Francisco Amaro-Gahete http://orcid.org/0000-0002-7207-9016
Rocío Cupeiro http://orcid.org/0000-0002-4119-0002
María Trinidad González Cejudo http://orcid.org/0000-0002-7871-7453
Patricia Virginia García Pérez http://orcid.org/0000-0003-4068-7292
Carlos Hernández-Martínez http://orcid.org/0000-0001-9116-9319
Raquel Sevilla-Lorente http://orcid.org/0000-0001-6396-7743
Alejandro De-la-O http://orcid.org/0000-0002-0614-4545
Alejandro López-Vázquez http://orcid.org/0009-0009-4545-6144
Marcos Molina-Fernandez http://orcid.org/0009-0008-9737-2666
Almudena Carneiro-Barrera http://orcid.org/0000-0002-3879-6468
Federico Garcia http://orcid.org/0000-0001-7611-781X
Alba Rodríguez-Nogales http://orcid.org/0000-0003-1927-0628
Julio Juan Gálvez Peralta http://orcid.org/0000-0001-6876-3782
Rafael Cabeza http://orcid.org/0000-0001-7999-1182
José L Martín-Rodríguez http://orcid.org/0000-0002-7538-4139
Araceli Muñoz-Garach http://orcid.org/0000-0002-1867-1158
Manuel Muñoz-Torres http://orcid.org/0000-0002-9645-3260
Idoia Labayen http://orcid.org/0000-0002-4334-3287
Jonatan R Ruiz http://orcid.org/0000-0002-7548-7138

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
