## [Reviewer comments · BMJ Open]

ARTICLE DETAILS

TITLE (PROVISIONAL)	Time-restricted eating and supervised exercise for improving hepatic steatosis and cardiometabolic health in adults with obesity: protocol for the TEMPUS randomized controlled trial
AUTHORS	Camacho-Cardenosa, Alba; Clavero-Jimeno, Antonio; Martin-Olmedo, Juan J.; Amaro-Gahete, Francisco; Cupeiro Coto, Rocio; Cejudo, María Trinidad González; García Pérez, Patricia Virginia; Hernández-Martínez, Carlos; Sevilla-Lorente, Raquel; De-la-O, Alejandro; López-Vázquez, Alejandro; Molina-Fernandez, Marcos; Carneiro-Barrera, Almudena; Garcia, Federico; Rodríguez-Nogales, Alba; Gálvez Peralta, Julio; Cabeza, Rafael; Martín-Rodríguez, José L.; Muñoz-Garach, Araceli; Muñoz Torres, Manuel; Labayen, Idoia; Ruiz, Jonatan

VERSION 1 – REVIEW

REVIEWER	Helda Tutunchi Tehran University of Medical Sciences
REVIEW RETURNED	14-Oct-2023

GENERAL COMMENTS	In my view, this study protocol is a well-designed randomized clinical trial. SPIRIT items are completely described and only the following minor comments should be addressed: 1) Please revise this sentence in introduction “However, energy-restricted approaches are still not a standard public health strategy due to their due to their lack of long-term sustainability and undesirable long-term metabolic adaptations, which certainly lead to weight regain even in highly motivated patients”.2) Please explain how confounding variables will be controlled?3) Please check the concordance of the tenses used throughout the text.4) At the end of discussion, please explain about the strengths and limitation of the study.
--

REVIEWER	YoonJu Song The Catholic University of Korea
REVIEW RETURNED	05-Dec-2023

GENERAL COMMENTS	This manuscript outlines a study protocol designed to investigate the effect of a 12-week time-restricted eating (TRE) intervention on hepatic fat and cardiometabolic health in obese adults. The study is well-designed, and the protocol is clearly articulated. However, the manuscript appears to follow a format resembling IRB documentation. Given that this is intended for publication in a scholarly journal, it is recommended to present the information in a more concise and summarized manner.
---

	The methods and analysis section is excessively long. I propose the following recommendations. 1) Please move '1.9. Sample size' and '1.10. Randomization and blinding' after '1.3. Recruitment and Screening'. 2) Please combine from '1.5. TRE intervention' to '1.7. Usual care group' into a consolidated '1.4. Intervention Description'. In addition, please shorten the length of '1.4.1. Supervised Exercise Interventions'. Figure 2 and Supplementary Figure 1&2 already presented key components of exercise intervention. Please condense this section for brevity. 3) Please separate '1.8.1. Outcome measures' into a distinct section labeled '1.6. Outcome measures'. The current 'Outcomes measures' section contains numerous items. For better organization, consider numbering the subcategories according to the order of items presented in Table 2. 4) Please provide additional details on Nutrition Education, including information on frequency, duration, and the mode of delivery (online or offline).
--	--

VERSION 1 – AUTHOR RESPONSE

Reviewer: 1

Dr. Helda Tutunchi, Tehran University of Medical Sciences

Comments to the Author:

In my view, this study protocol is a well-designed randomized clinical trial. SPIRIT items are completely described and only the following minor comments should be addressed:

1) Please revise this sentence in introduction "However, energy-restricted approaches are still not a standard public health strategy due to their due to their lack of long-term sustainability and undesirable long-term metabolic adaptations, which certainly lead to weight regain even in highly motivated patients".

Response: Thank you for your comments. Now, this sentence has been checked and improved: "However, energy-restricted approaches are still not a standard public health strategy due to their lack of long-term sustainability and undesirable metabolic adaptations, which certainly lead to weight regain even in highly motivated patients"

Comment

2) Please explain how confounding variables will be controlled?

Response: We have included a section explaining how the potential confounding variables will be assessed, and included an explanation in the statistical section.

Comment

3) Please check the concordance of the tenses used throughout the text.

Response: Done, comment appreciated.

Comment

4) At the end of discussion, please explain about the strengths and limitation of the study.

Response: Done, comment appreciated.

Reviewer: 2

Dr. YoonJu Song, The Catholic University of Korea

Comments to the Author:

This manuscript outlines a study protocol designed to investigate the effect of a 12-week time-restricted eating (TRE) intervention on hepatic fat and cardiometabolic health in obese adults. The study is well-designed, and the protocol is clearly articulated. However, the manuscript appears to follow a format resembling IRB documentation. Given that this is intended for publication in a

scholarly journal, it is recommended to present the information in a more concise and summarized manner.

Response: Thank you for your valuable feedback. In adherence to the CERT guidelines and to ensure comprehensive understanding of our study methodology by fellow researchers, it is imperative to provide detailed information in the manuscript. However, in response to the Reviewer's suggestion, certain sections have been concisely summarized to maintain clarity and brevity.

Comment

The methods and analysis section is excessively long. I propose the following recommendations.

1) Please move '1.9. Sample size' and '1.10. Randomization and blinding' after '1.3. Recruitment and Screening'.

Response: Done as suggested by the Reviewer.

Comment

2) Please combine from '1.5. TRE intervention' to '1.7. Usual care group' into a consolidated '1.4. Intervention Description'. In addition, please shorten the length of '1.4.1. Supervised Exercise Interventions'. Figure 2 and Supplementary Figure 1&2 already presented key components of exercise intervention. Please condense this section for brevity.

Response: Following the Reviewer's comment, we have summarized and combined these sections.

Comment

3) Please separate '1.8.1. Outcome measures' into a distinct section labeled '1.6. Outcome measures'. The current 'Outcomes measures' section contains numerous items. For better organization, consider numbering the subcategories according to the order of items presented in Table 2.

Response: We agree. We have taken into account the Reviewer's suggestion and have modified it according to Table 2.

Comment

4) Please provide additional details on Nutrition Education, including information on frequency, duration, and the mode of delivery (online or offline).

Response: We have included this relevant information on the section: "All participants will receive monthly in-person nutritional education session, lasting approximately 90 minutes..."